# Differential Evolution Based Layer-Wise Weight Pruning for Compressing Deep Neural Networks

**DOI:** 10.3390/s21030880

**Published:** 2021-01-28

**Authors:** Tao Wu, Xiaoyang Li, Deyun Zhou, Na Li, Jiao Shi

**Affiliations:** School of Electronics and Information, Northwestern Polytechnical University, 127 West Youyi Road, Xi’an 710072, China; tao_woe@mail.nwpu.edu.cn (T.W.); dyzhounpu@nwpu.edu.cn (D.Z.); linaflydream@mail.nwpu.edu.cn (N.L.); jiaoshi@nwpu.edu.cn (J.S.)

**Keywords:** neural network compression, weight pruning, differential evolution, sparse network

## Abstract

Deep neural networks have evolved significantly in the past decades and are now able to achieve better progression of sensor data. Nonetheless, most of the deep models verify the ruling maxim in deep learning—bigger is better—so they have very complex structures. As the models become more complex, the computational complexity and resource consumption of these deep models are increasing significantly, making them difficult to perform on resource-limited platforms, such as sensor platforms. In this paper, we observe that different layers often have different pruning requirements, and propose a differential evolutionary layer-wise weight pruning method. Firstly, the pruning sensitivity of each layer is analyzed, and then the network is compressed by iterating the weight pruning process. Unlike some other methods that deal with pruning ratio by greedy ways or statistical analysis, we establish an optimization model to find the optimal pruning sensitivity set for each layer. Differential evolution is an effective method based on population optimization which can be used to address this task. Furthermore, we adopt a strategy to recovery some of the removed connections to increase the capacity of the pruned model during the fine-tuning phase. The effectiveness of our method has been demonstrated in experimental studies. Our method compresses the number of weight parameters in LeNet-300-100, LeNet-5, AlexNet and VGG16 by 24×, 14×, 29× and 12×, respectively.

## 1. Introduction

Deep neural networks have significant performance in computer vision, speech recognition, natural language processing, etc. For sensor data processing, deep neural networks can also achieve good results. However, we must face up to some problems. Deep neural networks often require huge computing storage resources, while the resources of sensor devices are usually limited. Several investigate researches on deep neural networks, such as [1], provide a comprehensive analysis of important metrics for practical applications: accuracy, memory footprint, parameters, operations count, inference time and power consumption. With the development of deep neural networks, the number of network parameters increases rapidly. For example, in order to solve the ImageNet Large Scale Visual Recognition Challenge (ILSVRC), on the ImageNet dataset, AlexNet [2] has about 60 million parameters, while VGG [3] has about 150 million parameters. Even if some more efficient connections or modules are designed, such as residual network [4] or inception network [5], the size of parameter is reduced, but it is still huge. Deep neural networks with superior performance have deeper and wider architectures, which can result in expensive storage and computing costs. Especially for sensor devices or platforms, these deep neural networks are difficult to deploy as usual. Therefore, in order to reduce the size of the model, it is necessary to simplify the neural network architecture. Neural network pruning is a simple yet efficient method, which aims at removing some unimportant synapses and neurons to obtain sparse neural networks.

Neural network pruning is a traditional task that has received increasing interest with the development of neural networks. Especially, as deep neural network plays an increasing important role in practical applications, neural network pruning has become a highlight area of research, as it helps to achieve good working of deep neural networks on mobile or embedded devices. Moreover, neural network pruning is also a technique for designing or searching neural network structures. Back in 1990, LeCun et al. proposed Optimal Brain Damage (OBD) [6], which reduces the number of connections by finding the minimum active connections. In [7], Hassibi et al. used a method named Optimal Brain Surgeon (OBS) to remove the unimportant weights determined by the second-order derivative information. Compared to OBD, OBS has better pruning effectiveness, but the computational cost is more expensive. From these two methods onwards, many neural network pruning methods based on neurons or connections importance have been constantly proposed [8,9,10,11,12,13]. Moreover, some layer-wise methods [13,14,15] have been proposed for multilayer feed forward networks. For convolutional neural networks (CNNs), some structural pruning strategies, such as channel or filter pruning, have been proposed [16,17]. Among these structural pruning methods, they pay more to channel wise, kernel wise and inter kernel strides sparsity.

In model pruning, we not only want a sufficiently sparse network, but also want the network to retain as much of its original performance as possible. Most existing neural network weight pruning methods remove some unimportant weights or neurons by preset pruning ratio of each layer or global pruning ratio for the whole model. A simple experiment was conducted to verify the influence of pruning ratio per layer on pruning performance. In the experiment, we pruned the LeNet-300-100 that is a fully connected network with three layers, and the total pruning rate in the different pruning schemes is always 80%, but the pruning rate of each layer was different. The experimental results are shown in Table 1. The results revealed significant differences in the pruning results obtained by different pruning schemes. Therefore, We believe that this is not the best solution for model pruning, because different layers have different sensitivity to pruning, and this sensitivity is unpredictable. It is possible to promote pruning of the whole network, if each layer is finely pruned. Therefore, the challenge is to determine the pruning sensitivity of each layer. In addition, removing too many connections during pruning can cause accuracy loss, and iterative pruning (cycle pruning and fine-tuning) can be effective in maintaining the accuracy of the pruned model [12,14,18,19]. However, it is difficult to obtain high performance network with high sparsity by simple fine-tuning operation. Thus, we also need to improve the strategy of network fine-tuning after pruning to find a trade-off solution with better accuracy and sparsity.

To address the above problems, we first propose a differential evolutionary neural network compression framework (DENNC). In DENNC, we use a differential evolutionary algorithm to solve the problem of layer sensitivity to pruning. The mathematical representation of model weight pruning is an optimization problem with two terms, two of which correspond to sparsity and network accuracy. The decision variable for the above problem is the pruning mask, which can be determined by the sensitivity of each layer to pruning. After pruning the model with differential evolution, we will fine-tune the pruned model. We also loop the above two steps using an iterative pruning framework to obtain better solutions. It is worth noting that at the beginning of the model fine-tuning, we randomly recover a few of the removed connections to maintain the capacity of the model.

The remainder of this paper is organized as follows. In Section 2, we will review the background of proposed method and some related works. In the third section, the differential evolution framework for compressing neural network will be introduced in detail. We will conduct the experimental studies in Section 5. Finally, we will give concluding remarks of this paper.

## 2. Background and Related Work

### 2.1. Differential Evolution

Differential evolution (DE) [20,21,22] is one of the most well-known optimization algorithms in the field of continuous optimization because of its efficient performance. Compare to traditional evolutionary algorithms, DE generates new candidates through the weighted difference of the individuals in current population. More specifically, the weighted difference of several random individuals is added to current individual to generate new solution in mutation operation. After mutation, a crossover operation is performed on the original population and its corresponding mutant individuals. Finally, greedy selection is used to generate new population.

For mutation operations, there are five popular mutation strategies [22] in DE algorithms as below:DE/rand/1:
(1)$$v_{i}^{g}=x_{p1}^{g}+F\times \left(x_{p2}^{g}-x_{p3}^{g}\right)$$DE/best/1:
(2)$$v_{i}^{g}=x_{best}^{g}+F\times \left(x_{p1}^{g}-x_{p2}^{g}\right)$$DE/rand-to-best/1:
(3)$$v_{i}^{g}=x_{i}^{g}+F\times \left(x_{best}^{g}-x_{i}^{g}+x_{p1}^{g}-x_{p2}^{g}\right)$$DE/best/2:
(4)$$v_{i}^{g}=x_{best}^{g}+F\times \left(x_{p1}^{g}-x_{p2}^{g}+x_{p3}^{g}-x_{p4}^{g}\right)$$DE/rand/2:
(5)$$v_{i}^{g}=x_{p1}^{g}+F\times \left(x_{p2}^{g}-x_{p3}^{g}+x_{p4}^{g}-x_{p5}^{g}\right)$$
where $v_{i}^{g}$ is the mutant individual generated from the *i*-th individual $x_{i}^{g}$ in current population at the *g*-th generation. p1,p2,p3,p4,p5 are the random and mutually exclusive indices within the range [1, *n*], where *n* is the size of population. And $x_{best}^{g}$ is the best solution in the population of generation *g*. *F* is a positive scaling factor to scale the difference vector.

Crossover operation is performed on each population individual $x_{i}^{g}$ and its corresponding mutant individual $v_{i}^{g}$ to generate a trial individual:(6)$$u_{ij}^{g}=\left\{\begin{array}{c} \begin{array}{cc} v_{ij}^{g} & if(rand\leq CR) \end{array}\\ \begin{array}{cc} x_{ij}^{g} & otherwise \end{array} \end{array}\right.$$
where the crossover rate CR controls the probability of coping the gene from corresponding mutant individual. For each gene in a individual, we will perform above crossover operation to generate new solution.
(7)$$x_{i}^{g+1}=\left\{\begin{array}{c} \begin{array}{cc} u_{i}^{g} & \begin{array}{cc} if & f(u_{i}^{g}) \end{array}\leq f\left(x_{i}^{g}\right) \end{array}\\ \begin{array}{cc} x_{i}^{g} & otherwise \end{array} \end{array}\right.$$

Before selection operations, all of the above generated trial individuals are evaluated on the fitness function corresponding to optimization problem. Let $f(x_{i}^{g})$, $f(u_{i}^{g})$ be the fitness of individual $x_{i}^{g}$ and its corresponding trial individual $u_{i}^{g}$. Subsequently DE selects the best one between $x_{i}^{g}$ and $u_{i}^{g}$ to enter the new formed population, which can be shown as Equation (Equation 7).

### 2.2. Neural Network Compression

There is significant redundancy in deep neural networks which is mainly caused by over parameterization [23]. These over parameterized models require significant memory and computation resources, and are prone to model over-fitting problem. Therefore, it is necessary to compress the model by removing unimportant parameters to decrease the memory and computation cost. Generally, with the increasing of network compressing ratio, model computational cost is decreasing. For fully connected network, the ratio of computational cost is approximate to weight compression, the computational ratio is about half of compressing ratio for convolutional network [12]. Next, we will introduce some different researches about deep models compression method.

#### 2.2.1. Neural Network Pruning

Neural network pruning is a classic technique in the filed of model compression, and it can be traced back to the 90s in the 20th century [6,7]. In recent years, Han et al. [12,24] have proposed an iterative pruning method to remove small weights below a threshold. The method focuses on weight filtering to obtain architecturally sparse model. However, unless we design special hardware computing units, it is difficult to improve the inference speed of the compressed model. Therefore, some researchers design pruning against neurons to achieve the goal of speeding up model inference [16,25,26]. They define a number of neuron or filter importance metric indices to rank neurons or filters, and achieve model pruning by eliminating less useful neurons or filters.

#### 2.2.2. Parameter Quantization

Parameter quantization is another efficient method for compressing models. Recently, parameter quantization has two main research directions, one is weights sharing, the basic idea of which is that multiple network connections share a weight, and the other is weights reduction, i.e., weight represent with low bit. Han et al. [24] proposed a weight share method based on K-means clustering. The paper clusters the weight matrices of each layer into several clusters by K-means algorithm and represents the weight of the clusters by the cluster center value of each cluster. Chen et al. have designed a new network architecture, Hashnet [27]. A hash function is used to randomly group connection weights into hash buckets, with connections in each bucket sharing the same weight parameters. Dettmers [28] compressed the 32-bit gradient and activation values of to 8-bit, where 8-bit are used to represent floating point numbers. He also used a dynamically determination of the range of exponential and decimal bits to reduce errors. Literature [29] shows that when using a stochastic rounding, 16-bit fixed-point notation can significantly reduce the memory and floating point operations.

#### 2.2.3. Knowledge Distillation

Knowledge distillation can be regarded as a transfer learning method where knowledge from a trained large model (teacher model) is extracted and transferred to a tiny model (student model). Hinton et al. introduced the concept of knowledge in [30] by learning about soft goals. Moreover, they used a temperature parameter to control the soft level of the probability distribution. Romero et al. [31] introduce intermediate layer hits to improve the model. The core idea is to enable the student model to learn the intermediate representation from the teacher model. In [32], Yim et al. argue that the relationship between layers is a better representation of the knowledge than the model output. Therefore, they calculate the flow of solution procedure (FSP) matrix to represent the relationship between layers and transfer the FSP matrix of teacher model to student model.

## 3. Methodology

In this section, we introduce the proposed differential evolutionary neural network compression method in detail. The proposed method adopts an iterative pruning and fine-tuning framework, and the differential evolution is used to achieve neural network weight pruning. Moreover, we adjust the fine-tuning strategy in order to promote the model capacity, and a simple computational complexity analysis is given at the end of this section.

### 3.1. Preliminaries

We formally introduce the symbols and annotations for neural network weight pruning firstly. The deep neural network can be parameterized by $\left\{W^{\left(i\right)}\in R^{N_{i}\times N_{i+1}}\;\mathrm{or}\;W^{\left(i\right)}\in R^{K\times K\times N_{i}\times N_{i+1}},1\leq i<L\right\}$, $W^{\left(i\right)}$ denotes the matrix of connection weights of the *i*-th layer. $N_{i}$ and $N_{i+1}$ denote the number of input and output neuron (channel) of *i*-th layer, respectively. If the dimension of R is 2, it is a fully connection layer, otherwise it belongs to convolutional layer. *K* means the kernel size, *L* denotes the number of layers. We define F as the mask that can guide the process of pruning, and the detailed pruning operation for each layer can be written as follows:(8)$${\hat{W}}^{\left(i\right)}=F^{\left(i\right)}⊙W^{\left(i\right)},$$
where $\hat{W}$ denotes pruned connection weights, ⊙ indicates the Hadamard product operator. The pruning mask F can be constructed according to pruning strategy. For example, when we use threshold method to prune fully connection layers, the F can be calculated by
(9)$$F_{ij}=\left\{\begin{array}{cc} 0 & \mathrm{if}\;|W_{ij}|<\zeta^{\left(l\right)}\\ 1 & \mathrm{otherwise} \end{array}\right.$$
where $\zeta^{\left(l\right)}$ denotes the pruning threshold of *l*-th layer.

### 3.2. Framework of Proposed Method

We follow the iteration pruning framework [12] which divides the neural network pruning into two phases. In the first phase, the original dense network is pruned to sparse network. In the second phase, the sparse network from previous phase is fine-tuned to recovery model performance. It is well known that removing connections affects the model performance, especially when too many connections are removed, the performance of the model is bound to degrade. And fine-tuning the pruned model helps to restore the model performance. Therefore, it is necessary to perform proper fine-tuning operations after differential evolutionary pruning, which can lead to a better pruned model.

The detailed framework of DENNC is summarized in Figure 1. In general, a dense neural network is used as input, such as the first network in Figure 1. The iterative pruning, which implements pruning and fine-tuning alternately, is used to improve final performance of neural network compression. In each iteration, we adopt a mode of fine-tuning after pruning. In the first phase, we implement differential evolutionary layer-wise weight pruning to simplify the dense network to a sparse network (as shown in the second network Figure 1), in which the best individual in population is selected as layer-wise pruning solution to obtain the pruned network. In the second phase, we fine-tune the previous pruned network by updating the reserved weights and reestablishing few connections in order to promote the model capacity. A illustrated result is shown as the third network in Figure 1. The differential evolutionary neural network weight pruning and fine-tuning strategy will be described in detail bellow.

### 3.3. Differential Evolutionary Neural Network Weight Pruning

For hierarchical neural networks, different layers can usually extract different features, and thus the weight distribution of each layer is different. Taking a fully connected neural network as example, it is impossible for us to remove 90% connections of each layer without losing accuracy. On the one hand, if we prune too many connections, the compressed neural network will not perform as well as before. On the other hand, if we remove too few connections, we will not get a sufficiently streamlined network. Therefore, it is not reasonable to use a uniform pruning threshold for all layers. To address this problem, we use a differential evolutionary algorithm to autonomously optimize the pruning sensitivity of each layer. We design the objective function of differential evolution to directly respond to the pruning effect, and set the pruning sensitivity of each layer as the decision variable of the objective function.

Before establishing objective function to optimize pruning sensitivity, we first denote the error ratio of the neural network E(·) by
(10)$$E\left(W\right)=\frac{FP+FN}{TP+FP+TN+FN}{|}_{W},$$
where *W* denotes the current weight of the model, TP—True Positive—the number of observations correctly assigned to the positive class; TN—True Negative–the number of observations correctly assigned to the negative class; FP—False Positive—the number of observations assigned by the model to the positive class, which in reality belong to the negative class; FN—False Negative—the number of observations assigned by the model to the negative class, which actually belongs to the positive class. For neural network weight pruning, the evaluation of pruning level can be stated as
(11)$$\begin{array}{cc} \; & f={∥F∥}_{l_{0}}\\ \; & s.t.\;∥E\left(\hat{W}\right)-E\left(W\right)∥<\epsilon , \end{array}$$
where ϵ is a very small number. We can transfer the constrained optimization problem (11) to the unconstrained problem with a parameter λ, and the pruning problem can be rewritten as
(12)$$\begin{array}{cc} \min f & =\lambda {∥F∥}_{l_{0}}\;+\;∥E\left(\hat{W}\right)-E\left(W\right)∥\\ \; & =\lambda {∥F∥}_{l_{0}}\;+\;∥E(F⊙W)-E(W)∥, \end{array}$$
where F can be calculated by Equation (Equation 9). In order to balance the huge numerical difference between these two terms, we normalize the ${∥F∥}_{l_{0}}$ as
(13)$$\begin{array}{cc} \min f & =\lambda \frac{{∥F∥}_{l_{0}}}{\left|F\right|}\;+\;∥E\left(\hat{W}\right)-E\left(W\right)∥\\ \; & =\lambda \frac{{∥F∥}_{l_{0}}}{\left|F\right|}\;+\;∥E(F⊙W)-E(W)∥, \end{array}$$
where $\left|F\right|$ denotes the number of elements in F. From Equations (Equation 9) and (Equation 13), we know that the parameter ζ affects the result of pruning and is independent for each layer. Therefore, we can refer to ζ as the pruning sensitivity.

The differential evolutionary algorithms have significant advantages in solving the NP-hard problem. The problem (Equation 13) is clearly an NP-hard problem, which is difficult to be handled by general optimization methods due to $l_{0}$-norm. Therefore, we use a differential evolutionary algorithm to handle the above problem for the purpose of analyzing pruning sensitivity. In the differential evolutionary pruning sensitivity analysis (DEPS), we let ζ be the decision variable, $Z={\left(\zeta_{1},\zeta_{2},...,\zeta_{L}\right)}^{T}$, which measures the pruning sensitivity of each layer. According to Equation (Equation 13), we simplify it appropriately to create the fitness function as follow
(14)$$\begin{array}{cc} \min f & =\lambda \frac{{∥F∥}_{l_{0}}}{\left|F\right|}\;+\;∥E(F⊙W)-E(W)∥\\ \; & =\lambda \frac{{∥F∥}_{l_{0}}}{\left|F\right|}\;+\;∥E(F⊙W)-C∥\\ \; & \approx \lambda \frac{{∥F∥}_{l_{0}}}{\left|F\right|}\;+\;E\left(F⊙W\right) \end{array}$$
where *C* denotes a constant. In this fitness function, we are interested in the pruning effect and the accuracy of the model after pruning, rather than the change in accuracy, which increases the probability of getting a more accurate model. The pseudocode of DEPS is shown in Algorithm 1. After running DEPS, we can obtain the final pruning sensitivity *Z* and the pruning mask F can be calculated by Equation (Equation 9). Next, the model will be pruned with mask F by Equation (Equation 8).
**Algorithm 1:** Pseudocode of DEPS
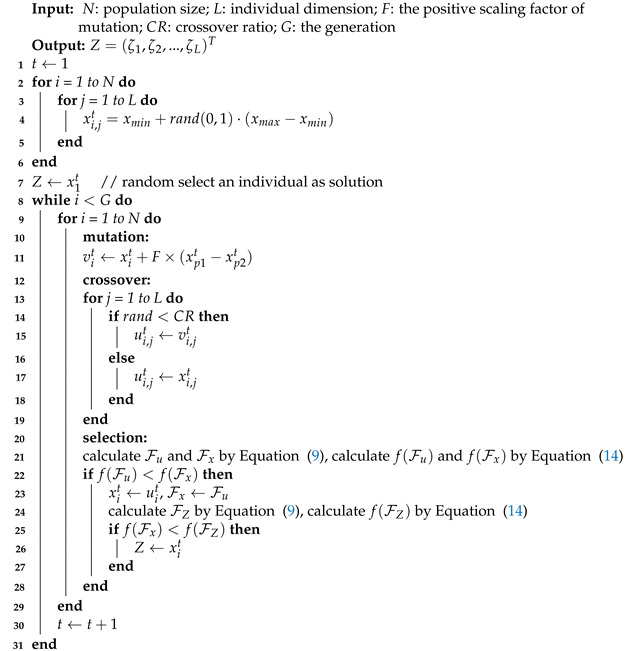


### 3.4. Fine-Tune Pruned Neural Network

Through the above pruning operation, we can obtain a sparse neural network, as shown in the second network in Figure 1. Generally, the pruned model is sparse but with some loss of performance. Therefore, it is necessary to compensate for the degradation in model accuracy caused by the directly removal of connections. Fine-tuning is a common strategy used in deep neural network training, which improves the performance of the model on specific problems by adjusting the pre-trained model. In our neural network pruning framework, we use the fine-tuning strategy to recover the model performance after the pruning operation.

In the model fine-tuning stage, we not only update the remaining weights, but also recover a few pruned connections, which is known as the recovery connections fine-tuning strategy. Updating the remaining weights helps to improve the accuracy of the model, while restoring pruned connections can promote the capacity og the model. During fine-tuning, we randomly recover some previously pruned connections and set random weights to them firstly, and then train the pruned neural network for a few epochs as usual. The detailed fine-tuning parameters are the same as those of the model training, except that the epochs are few.

### 3.5. Computational Complexity of DENNC

The computational complexity of DENNC model consists of two parts, the computational cost of DEPS for model pruning and the computational cost of fine-tuning pruned model. In DEPS, the main complexity comes from fitness evaluation. Assuming the number of weight parameters is *P*, the computational cost of fitness evaluation is O(P) for each individual. Therefore, for a population with *N* individuals, the computational cost is O(NP). The total computational cost of DEPS is O(NPG) because of generation *G*. Assuming the computational cost of neural network training for each epoch is O(T), the fine-tuning costs O(e·T) computations, *e* denotes the number of epoch in fine-tuning. Thence, the computational cost of DENNC in each cycle is O(NPG+e·T). If *K* stands for total number of iteration, the computational complexity of our method is O(K(NPG+e·T)).

## 4. Experimental Studies

In this section, we demonstrate the performance of DENNC with experimental studies. Firstly, we simply introduce datasets and experimental settings. Secondly, experimental results on LeNet, AlexNet and VGG16 are presented in detail. Lastly, we analyze some parameters and restoring connections fine-tuning strategies in the ablation studies.

### 4.1. Datasets and Experimental Settings

In experimental studies, we choose four different neural networks on two datasets, LeNet-300-100 and LeNet-5 [33] on MNIST, AlexNet [2] and VGG16 [3] on CIFAR10. A simple introduction about four models are shown as follow.

LeNet-300-100 is a fully connected neural network which has two hidden layers with 300 and 100 neurons, respectively.LeNet-5 is a simple convolutional neural network with two convolutional layers and three fully connected layers.AlexNet is a deep convolutional neural network, which includes five convolutional layers and three fully connected layers.VGG16 is a more deeper convolutional neural network with sixteen layers in total, in which thirteen convolutional layers and three fully connected layers are included.

Detailed network information, such as model type, accuracy and structure of each layer, is shown in the following Table 2. MNIST [33] has a training set of 60,000 examples, and a test set of 10,000 examples of handwritten digits. The images are centered in a 28 × 28 image. CIFAR10 [34] consists of 60,000 32 × 32 color images in 10 classes, with 6000 images per class. There are 50,000 training images and 10,000 test images in CIFAR10.

The comparison methods consist of naive cut [35], iterative pruning [12] and multi-objective neural network pruning (MONNP) [36]. Note all of these methods prune neural network with threshold optimization. Naive cut method removes the weights which are below the predefined global threshold. The computational complexity is O(P), which is the smallest in these methods, where *P* is the number of weight parameters. Iterative pruning method mentioned in [12] also prunes weights below predefined global threshold firstly, and then fine-tunes the pruned model, recursively runs the above two operations until the stop criterion is satisfied. When iteration number is *K*, the computational cost of iterative pruning is O(K(P+eT)), where *e* and *T* mean the epoch number in fine-tuning and the cost of fine-tuning for each epoch. The MONNP establishes a multi-objective neural network pruning model and uses multi-objective particle swarm optimization to handle the problem. Compared with proposed DENNC, the computational complexity is similar.

Note that all model used in experiment are trained by ourselves because we fail to obtain original model weights. Therefore there are some different results compared with that of reference papers.

### 4.2. Experimental Results

#### 4.2.1. Overall Results

Firstly, overall experimental results of proposed DENNC and comparison methods are shown in Table 3. In Table 3, we use five metrics to measure the effect of neural network pruning, which are the error of model, the number of weights, the percentage of pruned weights, the memory resource of model and compression ratio, respectively. It is clear that our method can efficiently prune these four models with a compressing ratio of 24.33, 14.47, 29.15 and 12.55, respectively. There is no doubt that the model obtained by our method requires minimal memory resource. From the perspective of model error, our DENNC performs best on LeNet-300-100 with smaller error, but on the other three models, the model error of DENNC is bigger than that of original model. Compared with the other methods, our method appears to be at a moderate level, and keeps acceptable errors. While it’s not doing very well on error, our method pruned the most weight parameters on all models. In the view of compressing ratio, the performance of our method is undoubtedly the best. In addition to our method, the iterative pruning method performs better. However, under the comprehensive consideration of error and compressing ratio, our method performs better than the iterative pruning method, because the iterative pruning method has bigger error on three models. In summary, the experimental results prove that our method can efficiently compress models and outperforms the comparison pruning methods.

Secondly, we analyze the performance of the proposed DENNC on LeNet-5 pruning task. Our DENNC belongs iterative pruning methods, and in each iteration, we pruning weights by the differential evolutionary layer-wise weight pruning method. So the fitness curves of the differential evolutionary layer-wise weight pruning under 100 generations are shown in Figure 2. There are five curves corresponding to the fitness changes of iteration 1, 2, 5, 15 and 30, respectively. We can clearly know that as the number of iterations increases, the overall fitness is decreasing. For example, the green line has smallest fitness and the corresponding iteration is the maximum of 30. It reveals that our iterative pruning is effective. For each curve, as evolutionary generation increases, the fitness is progressively decreasing in general, even if it remains unchanged in some generations. We can also observe that the range of variation for each curve is very limited, which indicates that the effect of a single pruning session is limited and that iterative pruning is necessary. The detailed iterative pruning effects with the number of iteration is shown in Figure 3. From the fitness curve, we can know that as the number of iterations increases, the fitness is decreasing in general. And in phases where the number of iterations is less than 5, fitness decreases faster, and as the number of iterations becomes larger, fitness decreases more slowly. The fitness is not always decreasing, sometimes the fitness may become slightly lager with the increasing of iteration. It is may be the result of our recovery connection fine-tuning strategy.

Next, we will analyze pruning results of each model in detail as follow.

#### 4.2.2. LeNet on MNIST

We firstly prune LeNet-300-100 and LeNet5 on MNIST. The detailed pruning results of each layer of these two neural networks are shown in Table 4 and Table 5, respectively.

For LeNet-300-100, we present pruning results of each layer in Table 4. From Table 4, we can know that the proposed method removes 95.89% weights in total. There are the most number of weights in the first layer, and the percentage of removed weights is also the highest. DENNC removes 97.08% weights of the first layer, and it is 85.75% weights of the whole network. For the second and third layers, 89.15% and 19.87% connections are pruned, respectively. In order to reveal pruning effect more intuitively, we plot histograms of weights distribution before and after pruning in Figure 4. Note that the number of bins in both histograms is 1000. From the figure, it is obvious that most of the small weights closed to zero are removed. we also notice huge difference between the two vertical axes, the maximum count in Figure 4a,b are 4500 and 90, respectively. Moreover, it leaves not only large weights but also a small amount of weights near zero after pruning.

For LeNet-5 model, detailed pruning results of each layer are shown in Table 5. Overall, 93.09% weights are pruned in the proposed DENNC. In LeNet-5, there are 92.44% weights in fully connected layers and 93.94% weights are removed of them. Among them, 96.03%, 93.06% and 19.41% connections of the three full connection layers were removed, respectively. For two convolutional layers, the proposed method prunes 10.18% and 77.49% weights, respectively. We also plot histograms of weights distribution before and after pruning for LeNet-5. In Figure 5, the weights approximate a normal distribution of zero mean, and most of the weights are in the range of [−0.4, 0.4] before pruning. After pruning, only a few small weights are retained, especially for the weights between [−0.2, 0.2]. From the above results, the proposed method is obviously effective for compressing LeNet-5.

#### 4.2.3. AlexNet on CIFAR10

AlexNet is a deep neural network which has five convolutional layers and three fully connected layers. We present detailed pruning results of each layer in Table 6. Totally speaking, 96.57% weights of AlexNet are pruned. For five convolutional layers, the proposed DENNC removes 24.85%, 28.28%, 35.89%, 55.54% and 27.04% weights, respectively. While overall pruning ratio for convolutional layers is about 40%, the weights of the convolutional layers are only 4.24% of the total. Most of the weights are in fully connected layers, especially in the first and second linear layers. For three fully connected layers, 99.03% connections are removed in total, where the pruning ratio of the first layer is 99.08%, the pruning ratio fo the second layer is 99.34% and 45.15% connections are removed in the third layer. Furthermore, we plot histograms of weights distribution before and after pruning in Figure 6, which aims to show pruning result more vividly. From Figure 6, we can know that most of the weights are in range [−0.15, 0.15] before pruning, and the count of weights which approximate zero is around $3\times 10^{4}$. After pruning, the maximum count in Figure 7b is around 800, which is very small compared with that of before pruning. It is obvious that the proposed method prunes a huge amount of weight. Corresponding to our pruning strategy, there is a small amount of weights close to zero after pruning. In a word, the proposed DENNC is work well for pruning AlexNet on CIFAR10.

#### 4.2.4. VGG16 on CIFAR10

VGG16 model is a very classical deep model with very a large amount of weight parameter. In this part, we prune VGG16 model on CIFAR10 and the results of each layer are shown in Table 7. Because of enough number of convolutional layers, the amount of convolutional layer weight parameters is not much different from that of the fully connected layers. From the total pruning result, 83.35% weights of convolutional layers are pruned, 98.73% connections of fully connected layer are removed and the proposed method prunes 92.03% weights in total. From the Table 7, we can know that most of the weights of convolutional are in layers of Conv9 to Conv13, and the pruning ratio of these layers is 75.43%, 93.40%, 96.57%, 95.17% and 96.98%, respectively. For fully connected layer, the pruning ratio of each layer is 94.60%, 99.36% and 87.19%, respectively. Moreover, we also plot histograms of weights distribution of VGG16 model in Figure 7. The figure shows that most of the weights are in range [−0.1, 0.1] before pruning, and the maximum count in Figure 7a is approximate to $1.5\times 10^{4}$. After pruning, the maximum count in Figure 7b is around 1000. Comparing Figure 7a,b, it is clear that a large amount of weight are removed especially for the weight which is close to zero. However, there are still few small weights which are close to zero, it corresponds to the strategy of recovery connection fine-tuning.

### 4.3. Ablation Studies

In order to analyze the sensitivity of balance parameter and the effectiveness of used strategy, we make ablation studies as follow. The ablation studies involve two experiments, parametric sensitivity analysis and fine-tuning strategy analysis.

#### 4.3.1. Parametric Sensitivity Analysis

According to definition above, the fitness function includes two terms, and these two terms are contradictory usually. Thence, the balance parameter λ would be influential for neural network pruning. We design following experiment for studying parametric sensitivity. In this experiment, we separately discuss the pruning effect in the case of λ equals to 0.1, 0.2, 0.4, 0.6, 0.8, 1, 2, 4, 6, 8 and 10. The detailed results are shown in Table 8 and Figure 8.

In Table 8, there are three indices to evaluate pruning performance. The fitness can provide an overall estimation for weight pruning, and the accuracy and sparsity measure two important characteristics of weight pruning, respectively. From the table, we can know that the parameter λ does affect the result of neural network weight pruning. Where we apply different λ, there are significant differences on these three indices especially on the accuracy and sparsity. And the fitness becomes larger with increasing parameter λ in general. In order to show the influence of the parameter λ more intuitively, we plot two curves about the accuracy and sparsity in Figure 8. In Figure 8, blue curve represents the accuracy and orange curve means the sparsity. It is clear that as the parameter λ increases, the accuracy is decreasing, but the sparsity is increasing. It is aligned with Equation (Equation 14), the larger the λ, the greater the weight of the sparse term, therefore the sparsity of pruned model is increasing with increased λ. Moreover, we can know that when λ is in the range [0, 1], the sparsity changes drastically from 0.86 to 0.94, while when λ is in the range [1, 10], the sparsity only increases from 0.94 to 0.98. And the changes of the accuracy is smoother in general. Thus, it can be seen that the sparsity is more sensitive to λ than the accuracy.

#### 4.3.2. Fine-Tuning Strategy Analysis

In proposed DENNC, we adjust the fine-tuning strategy to obtain better model capacity. In previous iterative pruning methods, fine-tuning is used to update remained weights. However, we use fine-tuning to update not only remained weights but also some removed weights. In this part, we will analyze the effectiveness of recovery connections fine-tuning strategy.

Firstly, we analyze the influence of recovery connections fine-tuning strategy for pruning fitness. As shown in Figure 9, blue histogram (DENNC) denotes pruning with recovery connections fine-tuning strategy and orange histogram (DENNC-N) represents pruning with normal fine-tuning strategy. The horizontal axis denotes iteration and vertical axis indicates fitness of pruned LeNet-300-100. From the figure, the fitness decreases with the increasing iteration, and the fitness of DENNC is less than that of DENNC-N in the most iteration.

Secondly, in order to explore the impact of fine-tuning strategy more intuitively, especially for model accuracy, we show two accuracy curves of the pruned models in Figure 10. In Figure 10, we use orange and blue curves represent the accuracy of pruned LeNet-300-100 with and without recovery connections fine-tuning strategy, respectively. We can know that the pruned model using recovery connections fine-tuning is alway has better accuracy. It is in accord with the purpose for which we designed this strategy. Therefore, we can infer that the recovery connections fine-tuning strategy is efficient to improve model accuracy.

Finally, we plot weight distribution of pruned LeNet-300-100 with and without recovery connection fine-tuning strategy in Figure 11 to analyze the difference between these two pruned model. From the figure, it is obvious that recovery connections fine-tuning strategy could reserve more small weights. Figure 11a shows that there is a certain number of weights near zero, but it does not exist in Figure 11b. Even though we assume that the smaller the weight, the less important the corresponding connection, it does not mean that small weights are useless. Sometimes small weights could bring more subtle features which is also important for improving model performance.

## 5. Conclusions and Future Works

In this paper, we proposed a differential evolutionary weight pruning method to compress neural networks. It is based on iterative pruning framework, consists of two phases, model pruning and model fine-tuning, respectively. In model pruning phase, we analyzes the pruning sensitivity of each layer by differential evolutionary approach, and then the sensitivity is used to guide the pruning process. In model fine-tuning phase, the connections that have been removed are also considered for proper recovery, which is able to improve model capacity. Experimental results demonstrate that our method can efficiently compress model, there is least 10× compressing ratio for each model, especially 29× compressing ratio in AlexNet. Compared with similar threshold pruning methods, out method performs better than these comparison methods overall. Moreover, we also do ablation studies for parametric sensitivity analysis and fine-tuning strategy analysis. Experiments show parameter λ is able to reflect the pruning result, and better model can be obtained by using fine-tuning strategy.

There are still some unresolved issues in this paper, for example, the time cost of the proposed method is expensive and with obtained sparse neural network it is also hard to accelerate inference. Therefore, in future work, we want to speed up the proposed method with parallel computing. Furthermore, we will also focus on neural network structural pruning with heuristic optimization.

## Figures and Tables

**Figure 1 sensors-21-00880-f001:**
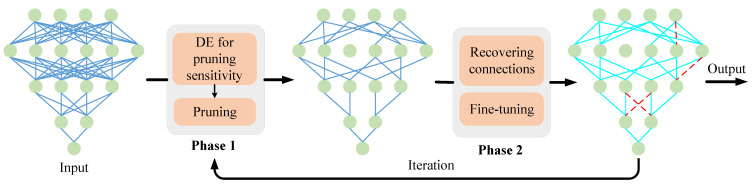
The iteration pruning and fine-tuning framework of differential evolutionary neural network compression. For input dense neural network, we analyze the layer’s pruning sensitivity by differential evolutionary and then pruning this network. Next, we fine-tune the previously pruned networks with the strategy of properly recovering pruned connections.The recovered connections is shown as red dotted line. Iterating through above operations until the stop criteria is satisfied.

**Figure 2 sensors-21-00880-f002:**
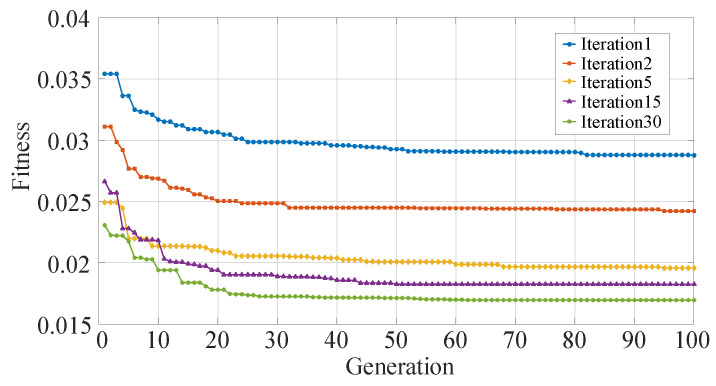
The fitness curves of differential evolutionary neural network compression framework (DENNC) under 100 generations on LeNet-5 pruning task.

**Figure 3 sensors-21-00880-f003:**
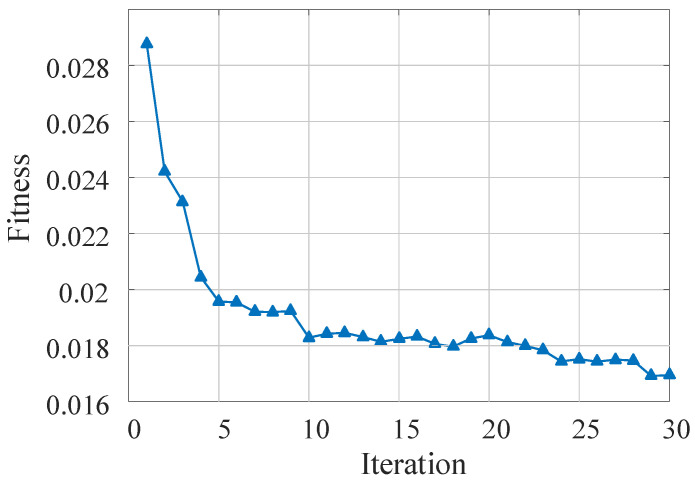
The fitness curve of DENNC under 30 iterations on LeNet-5 pruning.

**Figure 4 sensors-21-00880-f004:**
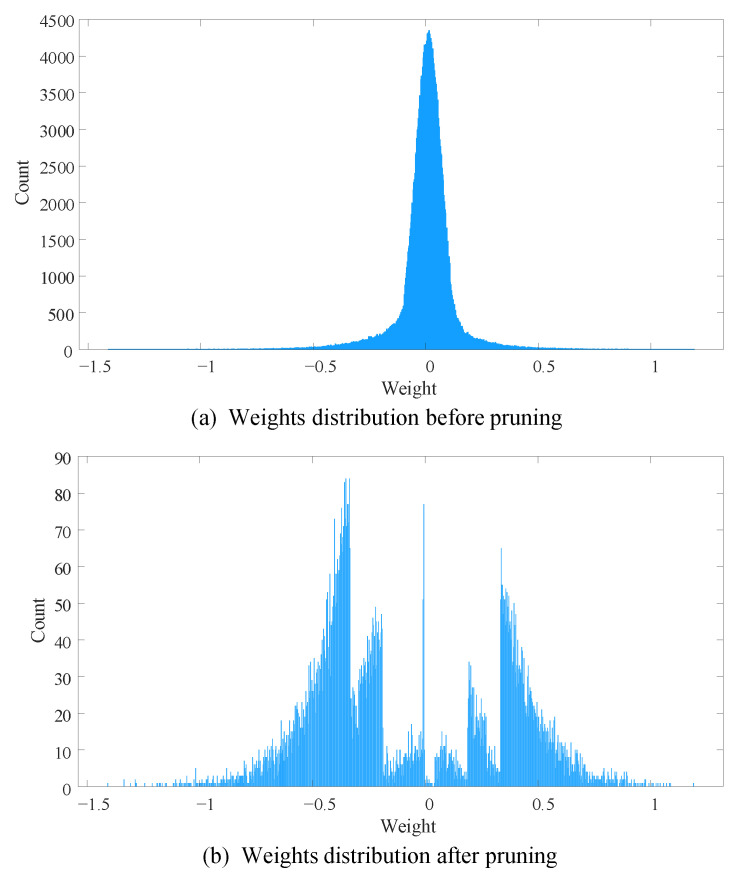
The weight distributions of LeNet-300-100 before and after pruning. The number of bins in both histograms is 1000.

**Figure 5 sensors-21-00880-f005:**
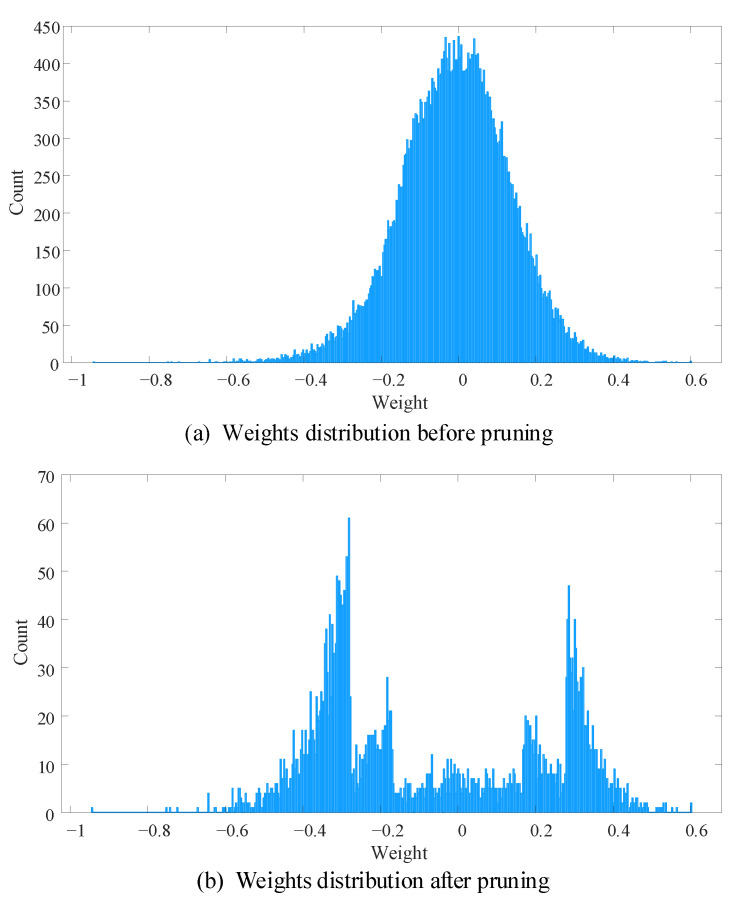
The weight distributions of LeNet5 before and after pruning. The number of bins in both histograms is 500.

**Figure 6 sensors-21-00880-f006:**
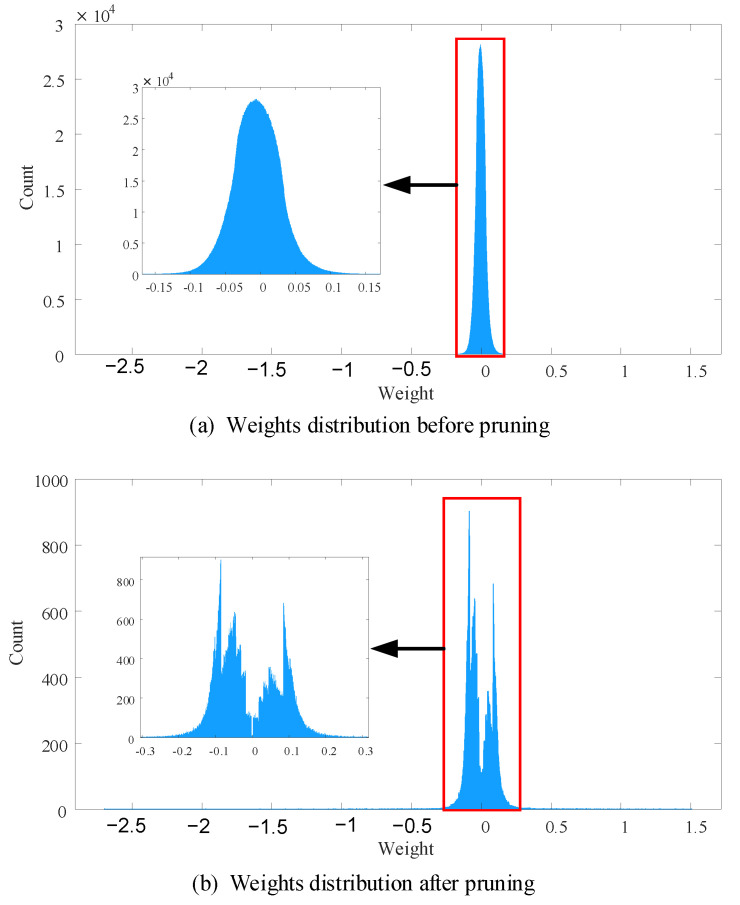
The weight distributions of AlexNet before and after pruning. The number of bins in both histograms is 10,000.

**Figure 7 sensors-21-00880-f007:**
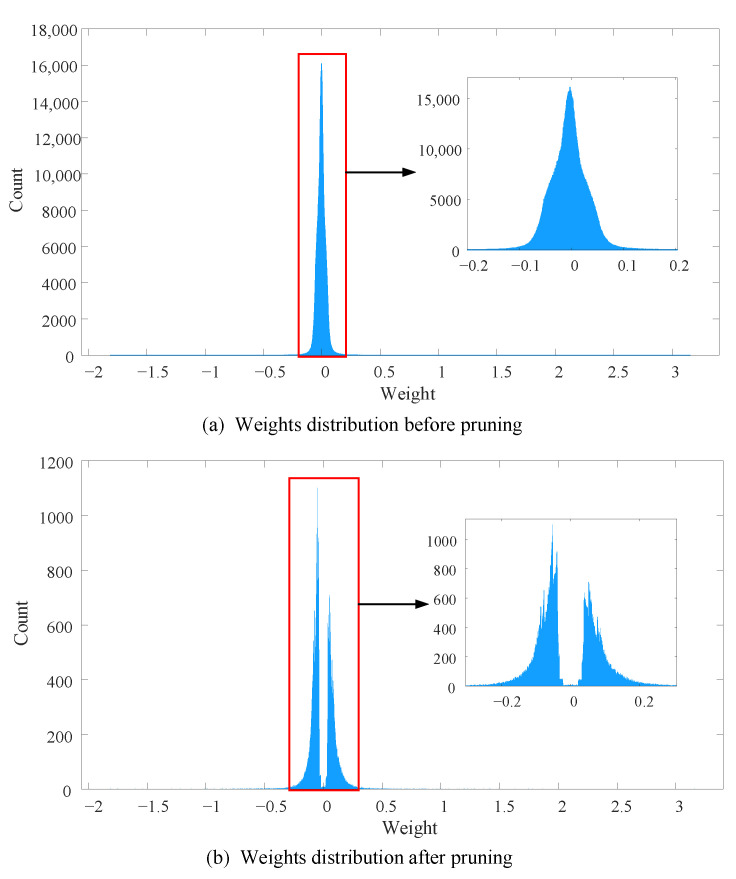
The weight distributions of VGG16 model before and after pruning. The number of bins in both histograms is 10,000.

**Figure 8 sensors-21-00880-f008:**
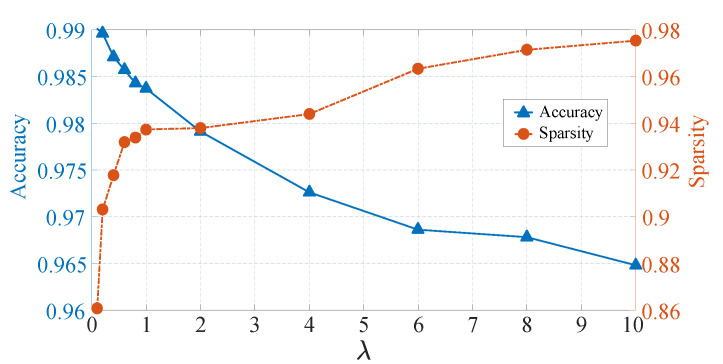
Changes in various indexes under different λ.

**Figure 9 sensors-21-00880-f009:**
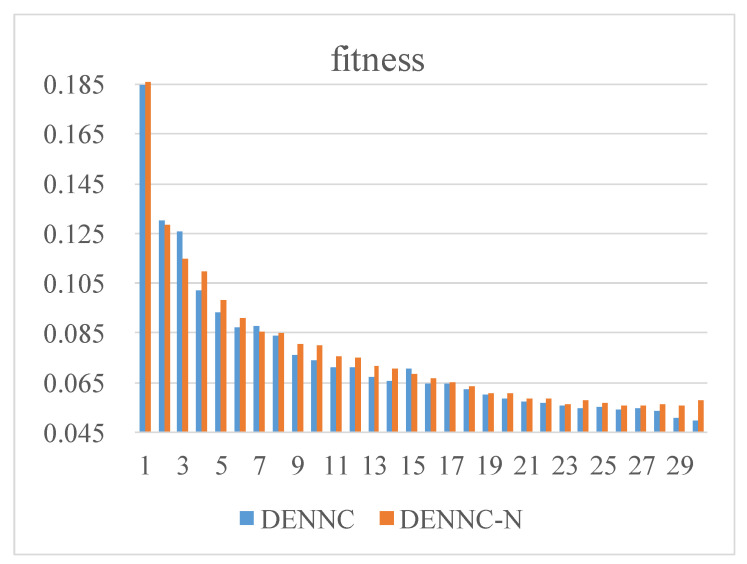
Changes in fitness with and without the pruning strategy of recovery connections for LeNet-300-100. The horizontal axis denotes iteration of DENNC, and the vertical axis is fitness.

**Figure 10 sensors-21-00880-f010:**
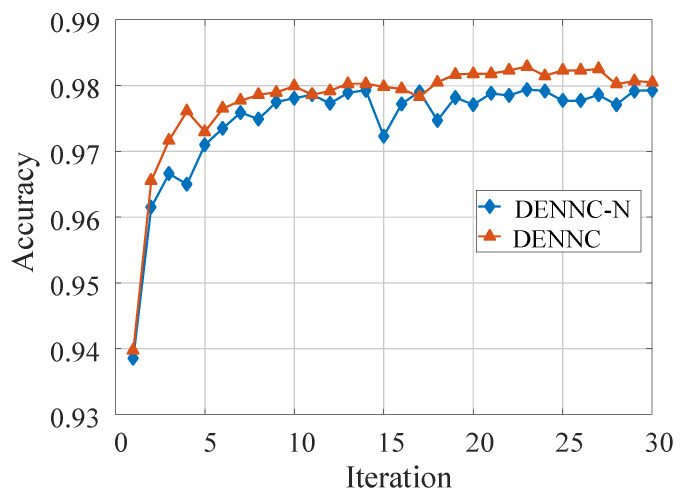
The accuracy of pruned LeNet-300-100 with and without recovery connections fine-tuning. The orange curve indicates the accuracy of pruned model with recovery connections fine-tuning, and the blue curve means the pruning result without recovery connections fine-tuning. The accuracy of DENNC is better than that of DENNC-N.

**Figure 11 sensors-21-00880-f011:**
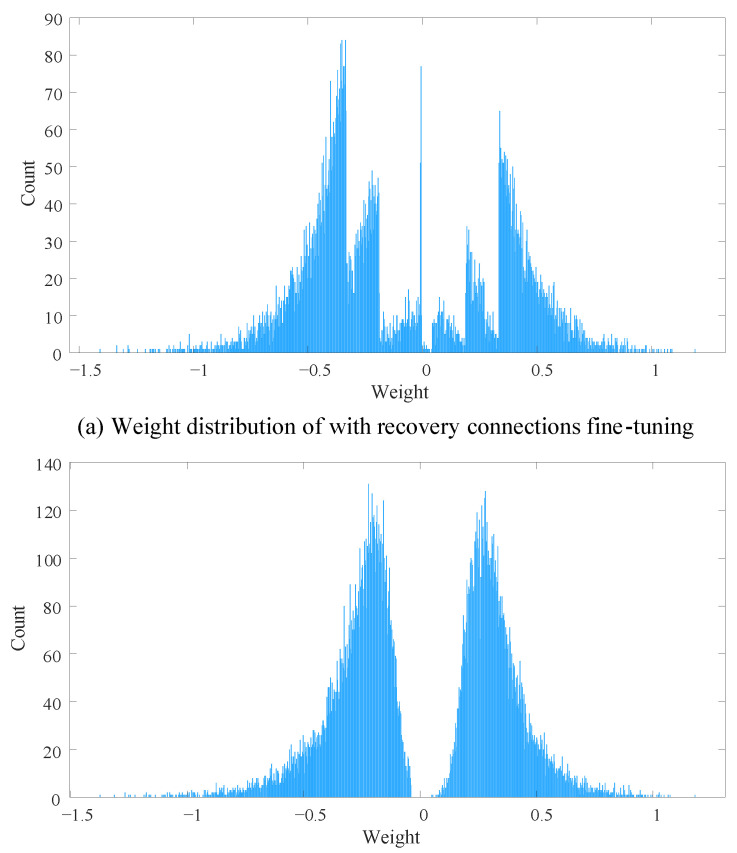
The weight distributions of LeNet-300-100 with and without recovery connections fine-tuning. The bins of these two histograms is 10,000.

**Table 1 sensors-21-00880-t001:** The simple test results on the sensitivity of the neural network layer to pruning. The test neural network is LeNet-300-100, and which is a fully connected network. There are different pruning ratio of each layer which are shown in the first three columns, and the last column is the accuracy of the pruned networks.

Pruning Ratio of Layer1	Pruning Ratio of Layer2	Pruning Ratio of Layer3	Accuracy
0.8	0.8	0.8	0.6701
0.82	0.68	0.1	0.9386
0.84	0.5	0.4	0.9294
0.86	0.35	0.3	0.9050
0.88	0.2	0.2	0.8764
0.90	0.05	0	0.8399

**Table 2 sensors-21-00880-t002:** Detailed neural networks information.

Model	LeNet-300-100	LeNet-5	AlexNet	VGG16
**Dataset**	MNIST	MNIST	CIFAR10	CIFAR10
**Type**	FCN	CNN	CNN	CNN
**Accuracy**	0.9774	0.9905	0.9004	0.8757
**structure**	(784,300)	8@(5,5)	24@(3,3)	16@(3,3)
	(300,100)	16@(5,5)	64@(5,5)	16@(3,3)
	(100,10)	(,120)	96@(3,3)	32@(3,3)
		(120,84)	96@(3,3)	32@(3,3)
		(84,10)	64@(5,5)	64@(3,3)
			(,1024)	64@(3,3)
			(1024,1024)	64@(3,3)
			(1024,10)	128@(3,3)
				128@(3,3)
				128@(3,3)
				128@(3,3)
				128@(3,3)
				128@(3,3)
				(,1024)
				(1024,1024)
				(1024,10)

**Table 3 sensors-21-00880-t003:** Overall results of proposed method and comparison methods. The bold numbers mean the best results.

Model	Method	Error %	# W	Pruned W %	Memory (M)	CR
LeNet-300-100	original	2.26	266,610	-	3.08	-
	Naivecut	2.84	106,937	59.89	1.46	2.49
	Iterative pruning	2.37	26,874	89.92	0.34	9.92
	MONNP	2.2	44,364	83.36	0.57	6.01
	DENNC	**2.07**	**10,962**	**95.89**	**0.14**	**24.33**
LeNet-5	original	0.95	45,278	-	0.55	-
	Naivecut	1.71	19,198	57.5	0.26	2.35
	Iterative pruning	1.72	3826	91.55	0.05	11.83
	MONNP	**0.91**	8503	81.22	0.11	5.32
	DENNC	1.41	**3444**	**93.09**	**0.04**	**14.47**
AlexNet	original	9.96	5,488,106	-	41.92	-
	Naivecut	11.66	1,153,600	78.98	9.57	4.76
	Iterative pruning	11.51	275,503	94.98	2.19	19.92
	MONNP	**11.41**	748,578	86.36	5.99	7.33
	DENNC	11.59	**192120**	**96.57**	**1.47**	**29.15**
VGG16	original	12.43	2,112,730	-	37.29	-
	Naivecut	**12.94**	972,278	53.98	18.83	2.17
	Iterative pruning	16.79	211,696	89.98	3.94	9.98
	MONNP	16.42	312,895	85.19	5.93	6.75
	DENNC	13.16	**168,489**	**92.03**	**3.08**	**12.55**

**Table 4 sensors-21-00880-t004:** The detailed pruning results of each layer for LeNet-300-100 on MNIST.

Layer	# Original *W*	# Remained *W*	Pruned *W* %
Linear1	235,500	6886	97.08
Linear2	30,100	3267	89.15
Linear3	1010	809	19.87
Total	266,610	10,962	95.89

**Table 5 sensors-21-00880-t005:** The detailed pruning results of each layer for LeNet5 on MNIST.

Layer	# Original *W*	# Remained *W*	Pruned *W* %
Conv1	208	187	10.18
Conv2	3216	724	77.49
Linear1	30,840	1225	96.03
Linear2	10,164	624	93.86
Linear3	850	685	19.41
Total	45,278	3444	93.09

**Table 6 sensors-21-00880-t006:** The detailed pruning results of each layer for AlexNet on CIFAR10.

Layer	# Original *W*	# Remained *W*	Pruned *W* %
Conv1	672	505	24.85
Conv2	38,464	27,586	28.28
Conv3	55,392	35,514	35.89
Conv4	83,040	36,918	55.54
Conv5	55,360	40,389	27.04
Linear1	4,195,328	38,709	99.08
Linear2	1,049,600	6876	99.34
Linear3	10,250	5623	45.15
Total	5,488,106	192,120	96.57

**Table 7 sensors-21-00880-t007:** The detailed pruning results of each layer for VGG16 model on CIFAR10.

Layer	# Original *W*	# Remained *W*	Pruned *W* %
Conv1	448	324	27.59
Conv2	2320	1268	45.33
Conv3	4640	3099	33.20
Conv4	9248	4057	56.13
Conv5	18,496	15,200	17.82
Conv6	36,928	22,606	38.78
Conv7	36,928	21,315	42.28
Conv8	73,856	22,788	69.15
Conv9	147,584	36,257	75.43
Conv10	147,584	9746	93.40
Conv11	147,584	5066	96.57
Conv12	147,584	7130	95.17
Conv13	147,584	4455	96.98
Linear1	132,096	7139	94.60
Linear2	1,049,600	6724	99.36
Linear3	10,250	1313	87.19
Total	2,112,730	168,489	92.03

**Table 8 sensors-21-00880-t008:** The pruning results of LeNet-5 under different parameters.

λ	Fitness	Accuracy	Sparsity
0.1	0.02382	0.9901	0.8608
0.2	0.02978	0.9896	0.9031
0.4	0.04582	0.9871	0.9177
0.6	0.05516	0.9857	0.9319
0.8	0.06866	0.9843	0.9338
1	0.0790	0.9837	0.9373
2	0.1451	0.9791	0.9379
4	0.2518	0.9726	0.9439
6	0.2516	0.9686	0.9633
8	0.2610	0.9678	0.9714
10	0.2822	0.9648	0.9753

## Data Availability

Not applicable.

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
