# Peer review of "Differential Evolution Based Layer-Wise Weight Pruning for Compressing Deep Neural Networks"

_sensors, 2021, doi:10.3390/s21030880_

Round 1
Reviewer 1 Report
The paper proposes a novel layer wise way pruning to compress deeper networks. I find the paper to be tackling an extremely important problem which tends to be very novel especially with the rise of models everywhere. The complexity of models may hinder their explainability and useless connections inside the neural networks may also be reason for increasing their maintainability in general. Keeping models simple without decreasing their performance is the aim of this paper as they develop a differential evolution to search for an optimal restructuring of the structure to To optimize its internal connections without losing its performance.
I find the paper to be relatively easy to follow and clear. I just could not wrap up my head around some of the decisions of the authors were making. For instance when they recover if you connections, it was not clear to me how this fits into their differential evolution process. Because the evolutionary algorithm they are proposing, does actually primarily remove connections, so I did not know how the fitness function would react to adding connections afterwards. Wouldn't this be interfering with the evolution?
Why would this be added afterwards and not part of the constraints to the search space?
Reviewer 2 Report
Summary:
This paper describes a pruning technique of deep neural network called layer-wise weight pruning. The authors proposed a method that provides a pruning mask with randomly generating the sparse matrix for pruning. They evaluate the method applying to well-known neural network models. They conclude the method maintains better fitness than the related works.
Comments:
- In the introduction, although the authors insist on computational difficulty in embedded environment (sensor devices?) , it is not discussed after the introduction. It is not understandable what are the main merits in your proposed method in the aspect of computational resources. Please consider to add discussions regarding how the proposed method overcomes the difficulties.
- In section 3.5, the complexity of the proposed method is discussed. However, it is not clear how the complexity is improved against the one of the related works such as the conventional pruning methods compared in the evaluation in section 4.2.1. It is better to discuss the comparisons of the complexities in the section 3.5.
- The authors should discuss the memory resource impact when we use the proposed method. The conclusion describes that the authors need to improve the amount of computations for the proposed method. However, there is no discussion about the complexity of memory resources. It is better to discuss how the compression of the network affects to the use of memory focusing on the implementation and its complexity.
Round 2
Reviewer 2 Report
I agree the responses from the authors and the corresponding revisions.
To support the result, it would be better to add a discussion about a tradeoff between the network compression and the computational cost in section 2. It will guide the readers to image how much the computational cost would rise.
